# Mechanism of partial agonism in AMPA-type glutamate receptors

Hector Salazar[1,2,*], Clarissa Eibl[1,2,*], Miriam Chebli[1,2] & Andrew Plested[1,2]

Neurotransmitters trigger synaptic currents by activating ligand-gated ion channel receptors. Whereas most neurotransmitters are efficacious agonists, molecules that activate receptors more weakly—partial agonists—also exist. Whether these partial agonists have weak activity because they stabilize less active forms, sustain active states for a lesser fraction of the time or both, remains an open question. Here we describe the crystal structure of an α-amino-3-hydroxy-5-methyl-4-isoxazolepropionate receptor (AMPAR) ligand binding domain (LBD) tetramer in complex with the partial agonist 5-fluorowillardiine (FW). We validate this structure, and others of different geometry, using engineered intersubunit bridges. We establish an inverse relation between the efficacy of an agonist and its promiscuity to drive the LBD layer into different conformations. These results suggest that partial agonists of the AMPAR are weak activators of the receptor because they stabilize multiple non-conducting conformations, indicating that agonism is a function of both the space and time domains.

[1] Leibniz-Institut für Molekulare Pharmakologie, Robert-Rössle-Strasse 10, 13125 Berlin, Germany. [2] Cluster of Excellence NeuroCure, Charité-Universitätsmedizin Berlin, Charitéplatz 1, 10117 Berlin, Germany. * These authors contributed equally to this work. Correspondence and requests for materials should be addressed to A.P. (email: plested@fmp-berlin.de).

The ionotropic glutamate receptor (iGluR) family of ligand-gated ion channels provides the majority of excitatory neurotransmission in the central nervous system[1]. At synapses throughout the nervous system, activation of AMPARs (α-amino-3-hydroxy-5-methyl-4-isoxazolepropionate receptors) plays an essential role in complex thought, as well as processes such as learning, memory formation and brain development. Their ubiquity has also led them to be implicated in neurodegenerative and psychiatric disorders[1–3].

Structures of the full-length AMPA receptor in complex with different ligands[4] provide important details about the architecture of this receptor family. AMPARs assemble of four subunits[5], each comprising a large extracellular amino-terminal domain (ATD) a ligand binding domain (LBD) that is connected to the ion channel forming transmembrane domain (TMD) and a carboxyl-terminal domain (CTD). Glutamate released from the presynaptic terminal binds to the LBD of postsynaptic AMPARs, causing rearrangements that result in ion-channel opening and transduction of electrical signals.

The LBD has a bi-lobate fold, built up by an upper D1 and a lower D2 domain, with the ligand bound between the two lobes. In the active configuration, the LBDs dimerize through interactions between D1 domains. Upon binding of glutamate, the two domains close and envelop the neurotransmitter. This motion provokes the separation of the D2 domains, connected to the ion channel, and presumably leads to the opening of the receptor gate[6]. Nevertheless, the geometry and dynamics of the activation mechanism of this receptor remains opaque, not least because in all structures of the full-length AMPA receptor with agonists to date, the ion pore was closed, despite several arrangements of the LBD layer being observed.

One of the most pressing questions concerning ligand activity at enzymes, receptors and drug targets[7–9] concerns the mechanisms by which full and partial agonists produce different amplitude responses at equally full occupancies of available binding sites. Partial agonism can be described by two distinct classical models. The Monod–Wyman–Changeux (MWC) model imagines that the partial agonist is weaker in shifting its target to the active state[10], producing different divisions of state occupancy over time. In contrast the Koshland–Nemethy–Filmer (KNF) model suggests that the receptors can undergo sequential non-concerted changes of the structure[11], meaning that different ligands, like partial agonists, induce distinct conformational states, and thus a division over spatial variables, such as the closure of a clamshell domain. However, these two models are extremes of more general models, which include hybrid intermediates[12], but that might still be too simplified because they do not explicitly include multiple conformations. Methods reporting both conformational space and activation at once offer a promising avenue to investigate how these distinct models relate to activity.

Structural studies on isolated AMPAR LBDs in complex with agonists (that is, glutamate and willardiines) indicated a correlation between the cleft closure and the agonist efficacy[13]. However, although mutations at the 'mouth' of the cleft reduce glutamate efficacy[14,15], this correlation is not entirely reflected in full-length structures[16,17]. Also, MD simulations[18], NMR experiments[19,20] and single-molecule FRET analysis[21] showed that partial agonists lead to a variety of clamshell conformations. Thus, the correlation observed between clamshell closure and efficacy is not consistent, speaking against clamshell closure as the sole determinant of the degree of channel activation. In the context of the tetramer, further degrees of freedom, afforded by different agonists, could explain the differences in efficacy.

In two recent studies, we identified two distinct conformations of the tetrameric LBD layer, associated with partial activation of

the AMPA receptor. The 'closed angle' conformation (CA) is trapped by a crosslink between subunits A and C at position A665C (ref. 22) and represents an activation intermediate obtained in subsaturating glutamate. A similar compact form was subsequently obtained for LBDs fully bound to glutamate[23]. Here, we present the crystal structure of the soluble GluA2 LBD bound to the partial agonist 5-fluorowillardiine (FW) in a different tetrameric arrangement, which we could confirm with functional trapping in full-length receptors. In this arrangement, subunits B and D are found in close apposition, distinct from previously published structures. Extending our study across four distinct trapping geometries, and seven bridges in all, we found an inverse relationship between the efficacy of the agonist and the number of conformations that the AMPA receptor could adopt. These data show that the efficacy of AMPA receptor agonists is a function of both time and conformational space.

## Results

**Structure of a LBD tetramer in a partially-active state.** Several full-length crystal and cryo-EM structures of iGluRs have been recently solved. Among them, two crystal structures of GluA2 bound to the partial agonist FW in combination with activating toxin and positive modulator RR2b (ref. 16) (Fig. 1a) and the (S)-5-Nitrowillardiine (NW) (ref. 17) (Fig. 1b) have been reported. Both structures have a closed pore, and are most likely inactive or pre-active states, consistent with the idea that partial agonists activate the receptor only for a fraction of the time.

To unravel the mechanism underlying partial agonism we sought to investigate the intermediate states of AMPA receptor activation by a combined structural and functional approach. Therefore, we aimed to elucidate the structure of a tetrameric soluble LBD (sLBD) in complex with the partial agonist. Screening across ~200 conditions of mutants and different agonists, we obtained a crystal of the sLBD in complex with FW (Fig. 1c,d) that diffracted to 1.23 Å resolution.

The asymmetric unit harbours one molecule with density for residues Gly1 to Gly264 and a well-defined positive difference electron density for FW as seen from the ligand-free calculated omit-map (Fig. 1c; Supplementary Fig. 1). This allowed us to position the FW with occupancy of one and a B-factor of 7.0 (compared with a mean B-factor of protein atoms within 4 Å of 6.9, Table 1) indicating that the site was saturated with ligand. Within the crystal packing each FW-bound sLBD (sLBD_FW) forms a back-to-back active dimer with a symmetry mate, which further builds up into a physiologically plausible tetramer (Fig. 1d). The D1 dimeric interface in the presented sLBD_FW structure was intact and essentially indistinguishable from that of the apo (ligand free, PDB 4u2p, RMSD 0.5 Å)[16], antagonist-bound (DNQX, PDB 4l17, RMSD 0.5 Å)[22] and agonist-bound LBDs (Glu RMSD 0.4 Å)[23] when aligning the C-alpha atoms of D1. Intact D1 interfaces indicate that the structure is not related to desensitization[6,24].

Although similar to the previously described closed angle (CA) (ref. 22) and the tight conformation structure[23], some unexpected features were evident (Fig. 1d). LBD movement is an obvious step in opening the channel[22,25,26], so we measured distances between the four LBD subunits in the sLBD_FW structure to gain insight into how FW promotes the partial activity of the AMPA receptor. The diagonal subunit distance was measured between the Cα atoms of residue Ala665 on subunits A and C. Interestingly, the A–C distance increases from the active, fully Glu-bound sLBD state (PDB 4yu0, 8.3 Å) to the resting ligand-free state (PDB 4u2p, 12.5 Å), mainly because of the open arrangement of the dimers (Supplementary Fig. 2). This A–C distance is similar in the new sLBD_FW structure (13.1 Å), but results from

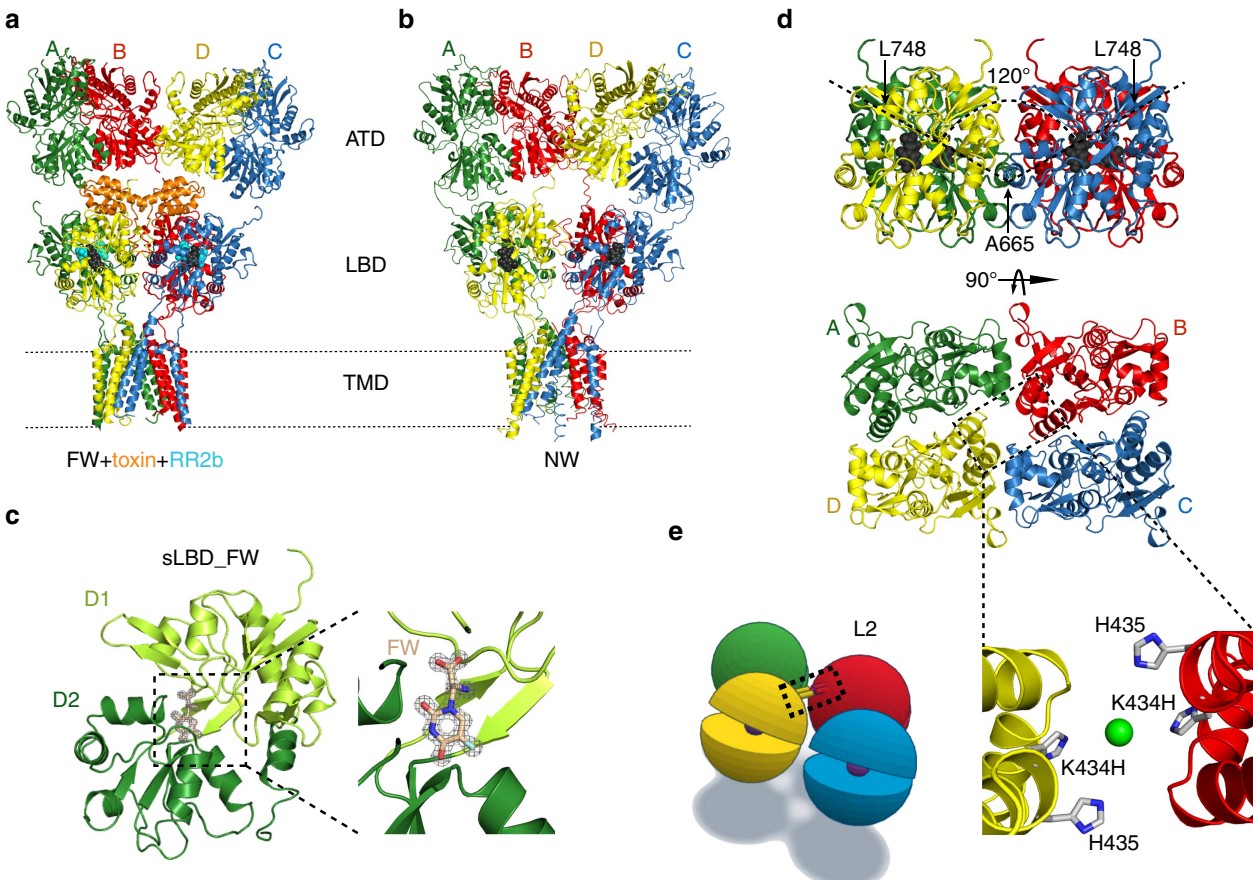

**Figure 1 | Structure of the GluA2 LBD tetramer bound by FW.** (**a**) Side view of the GluA2-toxin-(R,R)-2b-FW complex (PDB: 4u5c), with the subunits coloured as follows: A (green), B (red), C (blue) and D (yellow). The Con-Ikot-Ikot toxin is shown in orange and the agonist fluorowillardiine (FW) and the modulator (R,R)-2b are represented in black and cyan spheres, respectively. (**b**) The GluA2$_{NW}$ structure (PDB: 4u4f) represented as in **a**. The partial agonist nitrowillardiine is coloured in black spheres. (**c**) Crystal structure of the soluble LBD in complex with FW (sLBD_FW) at 1.23 Å resolution. Domain 1 (light green) and domain 2 (green) harbour the agonist binding site. The inset shows the positive difference electron density omit map around FW contoured at 2 sigma. (**d**) The upper panel shows the tetrameric sLBD_FW structure oriented as in the full-length structures presented in A and B. FW is shown as black spheres. The relative dimer orientation of 120° is measured by the angle between the centre of mass between Ala665 and Leu748 of subunits A and C. The lower panel shows the top view (turned 90° along the x-axis) onto the crystal structure of sLBD_FW, which forms a tetramer with its symmetry mates. (**e**) In the left panel a cartoon showing the localization of the mutant L2, which was used to validate the sLBD_FW structure is presented. According to the histidine mutation model based on our sLBD_FW structure (right panel) the L2 mutant should trap between the subunits B (red) and D (yellow). To probe this interface, a binding site for $Zn^{2+}$ was introduced by using the native histidine at position 435 and replacing residue K434 for a histidine. The fictive $Zn^{2+}$ ion is shown as a green sphere and was manually placed between the four histidines.

a shift of each dimer parallel to the interdimer interface and the plane of the membrane, and away from the overall axis of two-fold symmetry. The previously reported CA structure (PDB 4L17) was identified by functional measurements as an activation intermediate and is diagonally cross-linked by a disulfide-bond, which drove the shorter A-C distance (5.4 Å) and compact LBD arrangement. The CA structure got its name from its relatively closed interdimer angle of 113° relative to the antagonist-bound full-length structure (PDB 3kg2) (Supplementary Fig. 2). The angle is measured between the vectors created between Cα atoms of Ala665 on chain A and C and the Cα of Leu748 on the same subunits[23]. Considering the wide angle for full-length receptors in a putative apo state (165°) (ref. 27) and bound by glutamate proposed from CryoEM studies (181°) (ref. 25) (Supplementary Fig. 2), the 120° relative dimer orientation in the sLBD_FW structure (Fig. 1d) appears to be most similar to the geometry of an intermediate active state.

**Partial agonists allow a lateral shift of the LBD dimers.** According to our sLBD_FW crystal structure, the parallel shift of the dimers allows subunits B and D to approach to a surprisingly close extent (Fig. 1e). To validate this arrangement, we used a well-characterized metal ion trapping approach[22,23,28]. The mutant K434H (termed L2) is predicted to form a four-membered metal bridge with the native residue H435 (Fig. 1e). By analogy with previous work, if these histidines approach sufficiently to provide $Zn^{2+}$ coordination, changes of the receptor current during and after the partial agonist application in the presence of $Zn^{2+}$ might be detected. We reviewed recent crystal structures of the soluble LBD of GluA2, including: CA (DNQX-bound, 4l17), Tight (Glu-bound, 4yu0) and of the full-length GluA2 receptor in the ligand free state (Apo (4u2p) or bound to NW (4u4f), KA (4u2q), KA + RR2b + tox (4u5d), FW + RR2b (4u1y) and FW + RR2b + CII toxin (4u5c) for their potential to offer trapping at the L2 site. This analysis indicated that the possibility of the L2 bridge forming is unique to

**Table 1 | Data collection and refinement statistics.**

|  | GluA2 LBD bound to FW |
|---|---|
| *Data collection* | |
| Space group | P2$_1$2$_1$2 (#18) |
| Cell dimensions | |
| $a$, $b$, $c$ (Å) | 126.47, 44.42, 47.28, |
| $\alpha$, $\beta$, $\gamma$ (°) | 90.0, 90.0, 90.0 |
| Resolution (Å) | 47.28 − 1.23 (1.30 − 1.23)* |
| $R_{meas}$ | 6.9 (70.0) |
| $I/\sigma I$ | 14.5 (2.3) |
| Completeness (%) | 96.7 (92.8) |
| Redundancy | 4.07 |
|  | |
| *Refinement* | |
| Resolution (Å) | 32.37 − 1.23 |
| No. reflections | 75826 |
| $R_{work}/R_{free}$ | 12.1/14.7 |
| No. atoms | |
| Protein | 4174 |
| Ligand/ion | 22 |
| Water | 375 |
| *B*-factors | |
| Protein | 13.4 |
| Ligand/ion | 7.0 |
| Water | 26.9 |
| r.m.s. deviations | |
| Bond lengths (Å) | 0.012 |
| Bond angles (°) | 1.492 |

This structure was determined from one crystal.
*Values in parentheses are for highest-resolution shell.

the sLBD_FW structure (Supplementary Table 1). For example, the distance between the closest pose of the imidazole nitrogen atoms of residues His434 and His435 in subunits B & D is more than 18 Å in both the apo (ligand free) and fully active (glutamate-bound) states, which does not allow trapping (Fig. 2a,b). In our previous work on glutamate-bound LBDs, we discriminated between the two different arrangements based on crystal packing, which we called 'loose' and 'tight'[23]. In the 'loose' configuration, the residues of the mutant L2 are close enough to form a crosslink, but zinc exposure revealed only a small inhibition of receptor activation by glutamate, and in the absence of the desensitization-blocker, cyclothiazide (CTZ). To assess bridging by Zn$^{2+}$ over active states specifically, we added 100 µM CTZ and activated the L2 mutant with either glutamate or the partial agonists 5-fluorowillardiine (FW), 5-iodowillardiine (IW), willardiine (HW) and kainate (KA) (Fig. 2; Supplementary Fig. 3A). The response of the L2 mutant was not modified over a range of glutamate concentrations (10 µM–10 mM) in the presence of Zn$^{2+}$, indicating that the L2 crosslink cannot form when the receptor is glutamate-bound (Fig. 2c). In contrast, after zinc exposure in the presence of partial agonists, the active fraction was reduced (Fig. 2d). The most profound trapping was provoked by FW, which reduced the current elicited by 10 mM glutamate by 30% (Fig. 5), even though there was no discernible modification of the current during zinc exposure. The recovery of the current after trapping was relatively slow ($\tau = 290 \pm 30$ ms) indicating that this is one of the most stable crosslinks that we observe (Figs 2d and 5). Furthermore, we compared the extent of trapping from short (30 ms) to long time application (10 s) of Zn$^{2+}$, obtaining a further reduction in the active fraction of only 10% after 3 s of application, indicating that the reaction was already practically saturated after short intervals (Supplementary Fig. 3B,C). These striking results show that the sLBD_FW structure represents a unique arrangement of the LBDs, one that is available exclusively to partial agonists. Bridging is apparently

neutral with regard to efficacy, and cannot occur for glutamate-bound AMPA receptors (Fig. 2c).

**Partial agonists selectively support diagonal crosslinks.** In functional studies the A665C mutant could be selectively trapped at concentrations of glutamate close to the $EC_{50}$ ($\sim$500 µM), in the presence of the oxidizing agent copper-phenanthroline (CuPhen). This crosslink could be observed in the CA tetramer structure[22] (Fig. 3a). However, in the sLBD_FW structure, cysteine residues introduced at position 666 did not form a crosslink (their C-alphas being too distant at 17 Å). We therefore substituted three positions in the FG-loop (I664, A665 and V666) with cysteine (Fig. 3a). Each mutant was tested for its crosslinking potential with different partial agonists as FW, IW, HW and KA under oxidizing conditions (Fig. 3b). We used a quadruple barrel fast perfusion system that enables the fast application of each partial agonist in the presence of CuPhen, and allowed the partial agonist to dissociate whilst holding the disulfide bond with CuPhen before assessing the active fraction (Supplementary Fig. 4).

Perhaps surprisingly, given the sLBD_FW structure, we obtained trapping for the mutants I664C, A665C and V666C with all partial agonists FW, IW and KA (Fig. 3b; Supplementary Fig. 5A). However, HW failed to trap robustly at the I664C site (Active fraction $88 \pm 10\%$ $P = 0.0139$ versus WT see Fig. 5a). Receptors were better trapped at this site with the other partial agonists (active fraction range 77–82%), or at other sites (58–82% across 11 conditions).

**Promiscuous trapping by the weak partial agonist kainate.** Compact packing of the LBDs in the sLBD_FW structure produces interdimeric interfaces, including lateral subunit interfaces (between A & B and C & D) (Fig. 1d). To further validate this structure as a snapshot of an activation intermediate in full-length receptors, we used an already existing palette of engineered binding sites for Zn$^{2+}$ with distinct expectations of bridging by zinc, according to the sLBD_FW structure.

Two similar engineered sites, expected to crosslink in the sLBD_FW arrangement, occur between the D2 lobes of the subunits A & B or C & D. These sites are formed by the mutations D668H T672H K761H (termed T1) and D668H K765H D769 (termed HH)[23]. We also used a triple histidine mutant G437H, K439H, D456H (termed HHH), which was designed to bridge the D1 lobes[22] (for positions of the His-mutants, see Fig. 4a,c). In our FW structure, the distance between the residues across the lateral divide between dimers (again between subunits A & B or C & D) is larger than 20 Å, leading to the expectation that cross-linking would not be supported by FW (Fig. 4c, Supplementary Table 1).

The mutants T1 and HH trap when FW and KA are bound at high-concentrations of each partial agonist (Fig. 4; Supplementary Fig. 5B), in contrast to GluA2 WT (Fig. 4b). For T1 we observed a reduction of the active fraction of 24% and 23% for FW and KA, respectively; while the HH mutant displayed a reduction of the active fraction of 33% and 31% for FW and KA, respectively. These results are in agreement with our structural prediction (Fig. 5b and Supplementary Table 1). These observations contrast with measurements in the presence of low concentrations of glutamate, which protected against trapping[23] indicating that low occupancy by glutamate is not equivalent to saturating occupancy by a weak partial agonist (that is, kainate). Subsequently we extended this scan by analysing two additional partial agonists, IW and HW, which also showed a substantial reduction of the active fraction by 19% and 23% for IW and HW, respectively (Supplementary Fig. 5B). This observation confirms

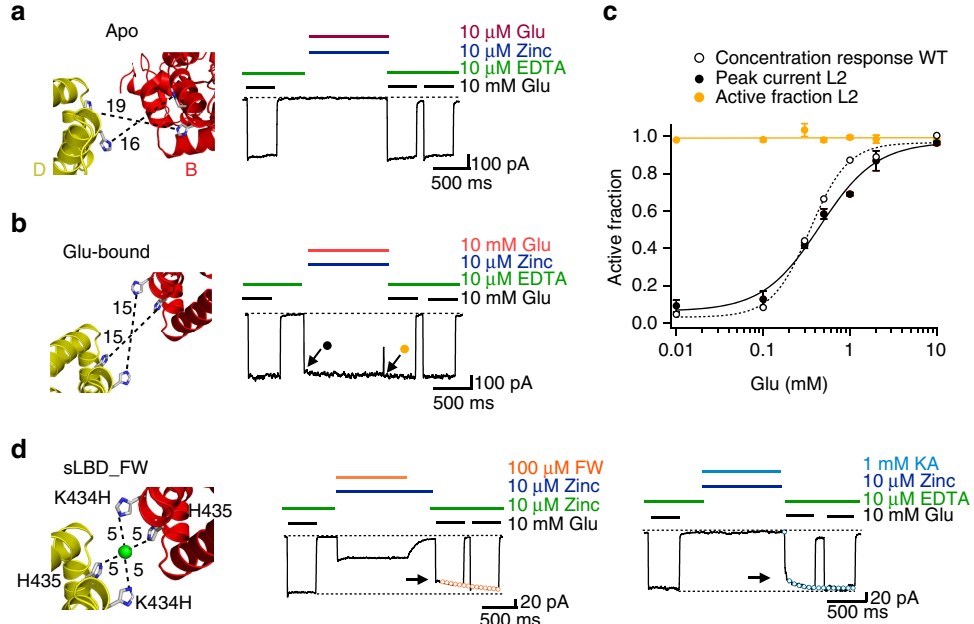

**Figure 2 | L2 traps selectively in presence of partial agonists but not in the Apo or the fully active states.** (**a**,**b**) Left panels show the L2 mutant models based on the apo (PDB, 4u2p) and the fully active (PDB, 4yu0) structures. The trapping histidines in these models are too distant (distances are measured between the imidazole N atoms of residues His434 and His435 in subunits B & D) to allow trapping. The right panels show patch clamp experiments at low concentrations of glutamate of 10 μM, testing the Apo state and 10 mM, testing the full activated state after the application of CuPhen. As predicted from the modelled histidines we do not observe changes in the active fraction. (**c**) Dose–response curves in Glutamate and 100 μM CTZ for WT GluA2 ($EC_{50} = 450 \pm 70$ μM) (open circles), the peak current, measured at the beginning of the application of $Zn^{2+}$ for the mutant L2 (closed circles) ($EC_{50} = 350 \pm 30$ μM). Zinc did not modify currents and the active fraction following zinc application (yellow circles) was fit by a linear function (c = 0.99; n = 4). (**d**) In the left panel, the L2 mutation modelled into the sLBD_FW structure could allow coordination of a fictive $Zn^{2+}$ ion (green sphere) by K434H and H435 from subunits B and D with minimal conformational change. Dashed lines indicate distances between fictive $Zn^{2+}$ and the imidazole N atoms. Trapping after the application of 100 μM FW and 10 μM $Zn^{2+}$ (left panel) and 1 mM KA and 10 μM $Zn^{2+}$ (right panel). Arrows indicate the reduction of the current after trapping in presence of $Zn^{2+}$. Open circles indicate double exponential fits of the recovery after trapping.

the attainment of these interfaces during the receptor activation with partial agonist.

Finally we used the HHH mutant as a negative control. When we applied FW, IW and HW at high concentration (1 mM for each partial agonist) in the presence of $Zn^{2+}$, we failed to observe any effect (active fraction range 99–100%; Fig. 4d; Supplementary Fig. 5b). Strikingly, the active fraction after the application of KA and $Zn^{2+}$ showed a reduction of $27 \pm 2\%$ (P = 0.0145 compared with wild-type) (Supplementary Fig. 5B). This observation suggests that the D1 lobes of subunits A & B and C & D would exclusively come into close proximity in the presence of KA, which is a poor agonist (efficacy ~10% in CTZ, Supplementary Table 2), compared with the other partial agonists used in this study, or to glutamate.

**Relationship between conformational space and agonism.** The striking failure of FW to promote crosslinking of the HHH mutant, in contrast with the robust trapping of this mutant in KA, suggests that the LBD tetramer can explore a larger conformational space with KA-bound than it can with FW occupying all four sites. However, knowing that the unbound LBDs are conformationally dynamic[28] immediately suggested that we might increase the conformational space available to the receptor by producing a mixed LBD layer with some subunits bound by FW and some empty. In other words, intermediate concentrations of FW should allow trapping by the HHH bridge, if a broad sampling of conformational space, and time division of state occupancy, are both contributing to low activity and therefore low efficacy of agonists. To investigate this hypothesis, we explore the reduction of the active fraction after the

application of $Zn^{2+}$ for the HHH mutant at different concentrations of FW. In low and high concentrations of FW there was no reduction in the active fraction (Fig. 6a). However, the relationship between the active fraction and log concentration followed an inverted bell shape, presenting a minimum at 3 μM (Fig. 6a,b). In agreement with our hypothesis, this result suggests that trapping at the HHH bridge occurred in states where FW was bound to some but not all sites. Inhibition was absent for WT (Fig. 6c).

If we compare the crosslinking profiles of KA, willardiines and glutamate, and we assume that each bridge corresponds to an exclusive arrangement (or set of arrangements) of the tetramer, we see that the worse the agonist, the more different conformations it can visit when the LBD layer is saturated (Fig. 7a). The lifetime of the bridges bore no relation to the extent of trapping, indicating that individual states were visited for different fractions of time (see Fig. 7b and Discussion). We also see that some bridges are visited in comparatively few tested conditions, whereas others (like the T1 bridge, which forms laterally between lobes 2) trap in almost any situation (Fig. 7c). Taken together, the poor ability of partial agonists to stably close the LBD clamshell produces increasingly profound conformational promiscuity. In turn, because some LBD tetramer conformations are ineffective at opening the channel, an increased propensity to visit different, inactive arrangements corresponds to worse agonist efficacy (summarized in Fig. 7d).

## Discussion

The first crystal structures of the GluA2 LBDs were solved nearly 20 years ago. Despite functional, biochemical, FRET, molecular

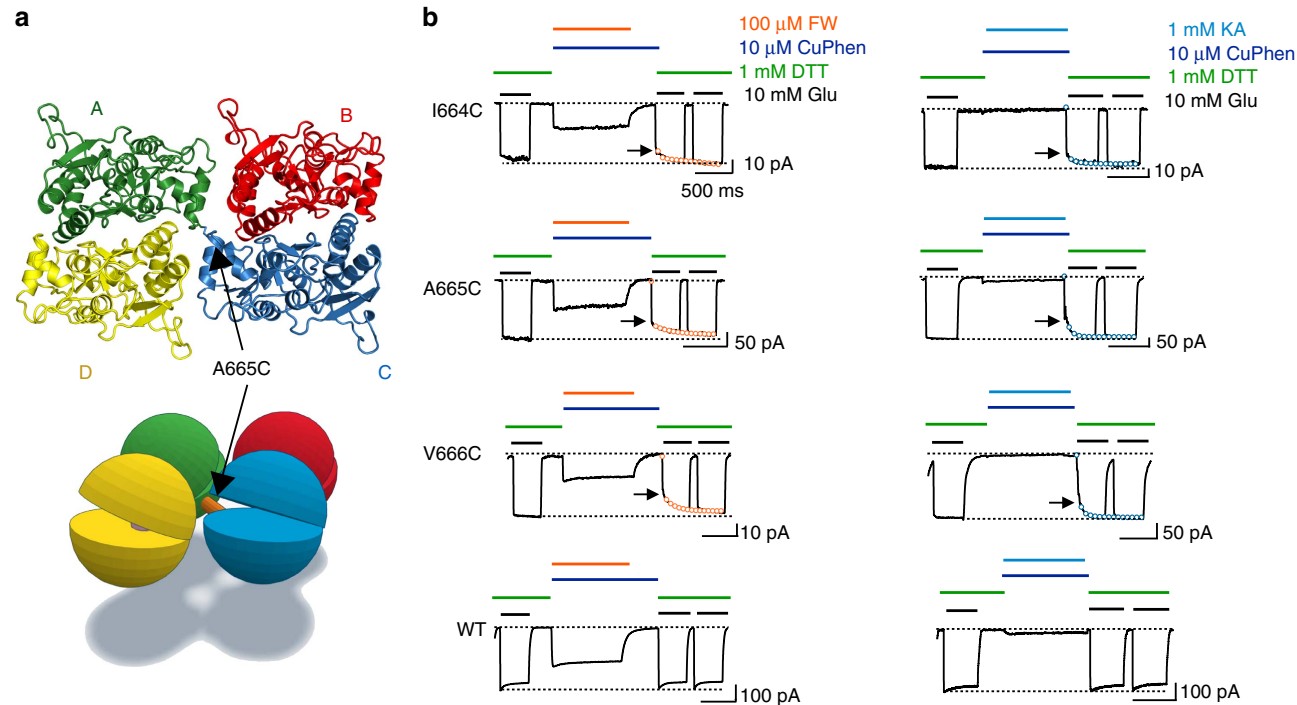

**Figure 3 | Partial agonists trap the A–C interface.** (**a**) Top view of the LBD layer of the CA structure[23]. Subunits A (green) and C (blue) are covalently linked by a disulfide bond formed between the introduced Cys665 on subunits A and C. The lower panel cartoon shows the localization of the position A665C. (**b**) Patch clamp experiments showing test pulses for cysteine mutations on positions 664-666. CTZ (100 μM) was present throughout the whole experiment. The recovery of the current in 10mM glutamate and DTT following trapping in the presence of CuPhen (10 μM) with 5-fluorowillardiine (FW) (right panel) or Kainate (KA) (left panel) was recorded. Arrows indicate a reduction of the current after trapping, which was observed for I664C, A665C and V666C but not for wild type (WT, bottom panel). Open circles indicate double exponential fits to the recovery after trapping. The time constants are summarized in Fig. 5. The difference between the active fraction after trapping in the presence of FW and KA for I664C, A665C and V666C compared with WT was significant for FW ($P < 0.005$) and KA ($P < 0.05$).

dynamics studies and the latest full-length structures of the glutamate receptor bound to agonists, antagonists and modulators, LBD movements during activation are not fully understood, perhaps because of their inherent complexity[16,29,30]. Nonetheless, there has been comparatively little testing of which conformations can be obtained by functional receptors undergoing normal activation in cell membranes[4]. By combining crystallographic studies with engineered binding sites for $Zn^{2+}$ or cysteine crosslinks, we could verify trapping of the full-length receptor in different functional states, assign these states to LBD tetramer geometries, and compare the properties of trapped receptors to the overall ensemble activated in a particular condition. Importantly, by confirming that mutations we made did not alter the channel opening equilibrium much in the absence of zinc, we ensured that we worked with receptors that were good mimics of wild-type receptors, (Supplementary Table 2). Unfortunately, receptor constructs heavily engineered for crystallization do not always obey this criterion. Trapping bridges can lock potassium channels and NMDA receptors into active, open channel states[31,32]. Perhaps surprisingly, given these observations, all the bridges that we have studied to date in the AMPA receptor, either inhibit activity when they coordinate zinc, or are neutral, as shown here for some partial agonists[23].

Our crystal structure of the soluble LBD in complex with FW (sLBD_FW) shares similarities with the full-length structures, like an intact D1 interface and a stabilized interface between subunits A&B and C&D, which was confirmed by cross linking experiments. However, a parallel shift of the dimers away from the pore and in the same plane as the membrane allows novel interdimer cross-links (L2) between subunits B&D. Further,

we could confirm that the interface HHH between the upper (D1) lobes of subunits A&B or C&D occurs only in certain circumstances, such as the presence of saturating KA or intermediate concentrations of better agonists like FW and glutamate.

Critically, the effects we see are not related to agonist affinity, because FW and HW have very different apparent affinities, despite their similar efficacies, and almost indistinguishable trapping profiles (Fig. 7). Recently, it was suggested that agonist affinity and efficacy might be very tightly related in nicotinic channels[33]. However, our data not only show that this is not the case in AMPA receptors, but provide an intuitive reason why not. Our measurements were made in the absence of desensitization and yet agonists with very different apparent affinities have essentially the same efficacies (compare HW and FW—94 fold difference in $EC_{50}$ and 1.1-fold difference in efficacy) (Supplementary Fig. 6), similar to a previous report[13]. Such decoupling, which may be inconsistent with a strict MWC interpretation, breaks down as soon as conformational space opens up in distinct ways for different ligands.

The mutants T1 and HH were both trapped in the presence of FW (Fig. 5) and the same interface is present in the NW crystal structure, with a distinct geometry. In the full-length crystal structures of FW and KA in the presence of the CII toxin the subunits are more separated. Interestingly, the L2 interface, between domains 1 of subunits B and D, which confirms the sLBD_FW structure is adopted in full-length, functional receptors, is not present in any of the recently published full-length crystal structures (FW + toxin + (R,R)2b) (ref. 29) and the

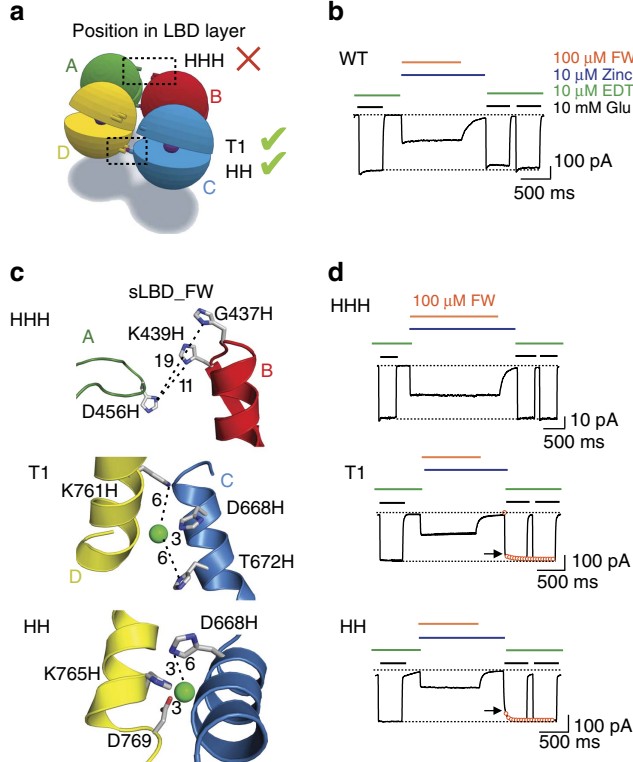

**Figure 4 | Failure to trap the D1 lateral interface is particular to saturating FW.** (**a**) Cartoon showing the localization of the mutants in the LBD layer. The red cross indicates that the HHH mutant is expected not to trap with FW, whereas the green ticks indicate proximity consistent with trapping. (**b**) Wild-type receptors did not show a decrease in the active fraction after the application of $Zn^{2+}$ and FW. (**c**) The mutated residues are depicted in the sLBD_FW structure. Expected distances between the histidine imidazole N-atoms and fictive $Zn^{2+}$ ions (green spheres) are denoted by dotted lines and expressed in Ångstroms. (**d**) Patch-clamp experiment showing the response of the HHH mutant, which was not modified in saturating FW. The difference between the active fraction after trapping in the presence of FW versus WT was not significant ($P = 0.5436$). The difference between the active fractions after the trapping of T1 and HH was induced by FW compared with WT was significant ($P < 0.009$). Arrows indicate the reduction of the current after trapping. The recovery of current in 10mM glutamate and EDTA following trapping (fit with double exponential function, circles) showed a common fast component, corresponding to the exchange of solution, and a slow component.

NW (ref. 30) (Fig. 1a,b). Whether the absence to date of this conformation in full-length structures is because of a crystallographic bias of catching one state out of many LBD arrangements, or that it represents a more active conformation whilst the channel in the structures is closed, will require further work.

We do not imagine that we have explored the full complement of arrangements that are available to the LBD tetramer, even though we have trapped 4 distinct geometries. Rather, it is fascinating that the pattern of different trapping geometries is so rich given the stereochemical limits of disulfide bonds and engineered metal bridges. These approaches can only detect proximity in the sub-nanometre range, and it is clear that open angles between LBD dimers are readily available[30,32] (although their functional consequences remain unclear). When we take a step backwards and note that all measurements here were made in the presence of CTZ to reduce intradimer mobility related to desensitization, wild-type receptors in synapses likely possess a

truly mind-boggling array of degrees of freedom, which, given the inhibitory nature of almost all bridges, mostly seem to serve the object of reducing activity.

Given that multiple subconductance states are available to the AMPA receptor pore, it seems that time and space division of activity is a general property that propagates throughout the receptor. On the other hand, in Cys-loop receptors, single channel measurements show that partial agonists activate the same conductance states of the receptor but with a decreased open probability[9]. The same scenario is observed in NMDA receptors[34], suggesting that the opening of the gate is an all or nothing event[10], and that partial agonism is a purely time domain process. These observations are consistent with the MWC idea that the same conformations (at least, the open or closed channel) are occupied for different fractions of the time according to agonist efficacy. If MWC applied to AMPA receptors, we would expect the same trapping profiles, independent of each agonist, but perhaps with the maximum extent of trapping developing over different time periods, corresponding to frequent or infrequent visits to a given state. On the other hand, if a pure KNF interpretation were correct, we might expect to see individual arrangements exclusively trapped, for different agonists. However, neither idea is consistent with our observation of overlapping sets of arrangements being visited.

Instead, a more general model that includes equilibrium between multiple arrangements[12] is needed to describe our observations. A broad conformational ensemble may result from concerted intermediate states, but in the case of the AMPA receptor, seems more likely to descend from independent transitions of individual subunits. Numerous observations support a relation between distinct agonist-dependent conformations of individual subunits and AMPA receptor activity. Consistent with the multiple conformations that the lobes can adopt in presence of partial agonists[20], twisting motions and other degrees of freedom have been reported[35–37]. Also small changes in the side chain orientations of the D2 lobes in GluA3 receptors bound to partial agonist, might be correlated to complex modal behaviour of the channel conductance[38,39]. Whether kainate and willardiines induce substantial differences in LBD dynamics remains a complex question[35,40], but here we show that they do induce distinct conformations of the LBD tetramer layer. Despite considerable study, teasing out a causal relationship between activity and subunit conformation in full-length receptors has not been trivial to date. Future work may investigate how manipulations at the level of individual subunits with known conformational effects might propagate into altered tetramer activity.

Given published data[13], we had expected to find a broad range of efficacies for willardiine ligands, but in our hands the range was narrow (30–50% between IW and HW, the largest and smallest substituents on the 5-position of the willardiine structure). Indeed, some differences in efficacy are apparent between expression systems and neurons, perhaps because of differential glycosylation, the use of particular mutants and auxiliary subunits. Agonism is in any case very sensitive to geometry of the LBDs and so the cause of these discrepancies might be hard to ascertain. Given that we saw a narrow range of efficacies, it was no surprise that the trapping profiles of different willardiine partial agonists were all similar- except that willardiine protected against trapping on the I664C bridge, consistent with a marginally higher efficacy and chiming with the protective effect of saturating glutamate against formation of the A665C bridge[22]. Consistent with a variable partition over distinct conformations of the LBD tetramer in time, we saw little correlation between the stability of a given trapped arrangement (inferred from the time constant of the loss of trapping in DTT or EDTA) and the extent

**a**

| CuPhen/DTT | FW | | | IW | | | HW | | | KA | | | |
|---|---|---|---|---|---|---|---|---|---|---|---|---|---|
| | Active fraction (%) | $\tau_{fast}$ (ms) | $\tau_{slow}$ (ms) | Active fraction (%) | $\tau_{fast}$ (ms) | $\tau_{slow}$ (ms) | Active fraction (%) | $\tau_{fast}$ (ms) | $\tau_{slow}$ (ms) | Active fraction (%) | $\tau_{fast}$ (ms) | $\tau_{slow}$ (ms) | |
| WT | 99 ± 2 | 3 ± 2 | – | 95 ± 9 | 2 ± 1 | – | 97 ± 1 | 4 ± 1 | – | 101 ± 3 | 5 ± 1 | – | |
| I664C | 82 ± 2 | 4 ± 1 | 208 ± 30 | 77 ± 2 | 2 ± 0.5 | 289 ± 53 | 88 ± 10 | 2 ± 1 | 133 ± 7 | 79 ± 2 | 2 ± 0.5 | 182 ± 50 | |
| A665C | 72 ± 1 | 4 ± 1 | 370 ± 90 | 74 ± 3 | 4 ± 1 | 197 ± 70 | 80 ± 3 | 2 ± 0.5 | 179 ± 38 | 73 ± 1 | 3 ± 1 | 154 ± 11 | |
| V666C | 58 ± 3 | 5 ± 3 | 219 ± 30 | 61 ± 8 | 6 ± 3 | 278 ± 82 | 66 ± 8 | 2.5 ± 0.5 | 168 ± 11 | 62 ± 3 | 8 ± 4 | 193 ± 23 | |

**b**

| Zn/EDTA | FW | | | IW | | | HW | | | KA | | | |
|---|---|---|---|---|---|---|---|---|---|---|---|---|---|
| WT | 98 ± 2 | 4 ± 1 | – | 94 ± 4 | 3 ± 2 | – | 97 ± 8 | 5 ± 1 | – | 98 ± 1 | 4 ± 1 | – | |
| T1 | 75 ± 3 | 7 ± 2 | 224 ± 47 | 80 ± 4 | 4 ± 2 | 239 ± 63 | 85 ± 2 | 9 ± 2 | 119 ± 24 | 76 ± 1 | 5 ± 1 | 101 ± 18 | |
| HH | 66 ± 3 | 7 ± 1 | 163 ± 37 | 59 ± 3 | 9 ± 3 | 97 ± 10 | 63 ± 2 | 7 ± 1 | 156 ± 55 | 68 ± 3 | 8 ± 1 | 163 ± 37 | |
| HHH | 100 ± 2 | – | – | 99 ± 2 | – | – | 99 ± 5 | – | – | 73 ± 2 | 7 ± 2 | 138 ± 19 | |
| L2 | 70 ± 3 | 7 ± 1 | 275 ± 26 | 78 ± 2 | 10 ± 2 | 185 ± 25 | 78 ± 2 | 2 ± 1 | 192 ± 3 | 75 ± 3 | 4 ± 3 | 198 ± 16 | |

**Figure 5 | Summary of current recovery after trapping. (a)** The reduction of the active fraction and the time constants obtained from the double exponential fitting to the current during the recuperation after trapping ($\tau_{fast}$ and $\tau_{slow}$) for cysteine mutants (for their locations, see Fig. 3). The fast time constant corresponds to glutamate activation, and the slow component is assumed to correspond to the breaking of the trapping bridge. **(b)** Equivalent data for histidine mutants (for their locations, see Figs 1 and 4). The graphs at the base of each column show the active fraction from all of the mutants versus the different partial agonist. The row end graphs show the individual mutants versus the different partial agonists. All errors are s.e.m., and for all data columns, $n = 5$ patches.

of trapping, which is a convolution of the geometry and the occupancy (Fig. 7). Therefore, equal time occupancies of given arrangements are incompatible with our observations.

In summary, by validating LBD trapping mutants, we found that the less efficacious the agonist is, the more distinct conformations the AMPA receptor can visit. In other words, partial agonists are bad at activating the channel because they allow the LBD to sample conformations that are unproductive. This observation suggests that partial agonism not only occurs by the limited ability of a given ligand to close the LBD, but also because of the (possibly consequent) reduced ability to constrain LBD movement within the tetramer.

## Methods

**Protein expression and purification.** Rat GluA2 ligand binding domain (S1S2 fusion) in pET22b vector was kindly provided by E. Gouaux. The V666C mutation was introduced by Overlap mutagenesis and sequence verified by DNA sequencing. The protein was expressed in *Escherichia coli* Origami B(DE3) cells with an N-terminal His$_8$-tag followed by a trypsin and thrombin cleavage site. Protein expression was induced with 100 µM IPTG at an OD$_{600}$ of 0.9. After 21 h at 20 °C, the cells were harvested by centrifugation at 12,000 g for 20 min, and lysed by the Avestin EmulsiFlex homogenizer in 20 mM Tris HCl, pH 8.0, 150 mM NaCl, 5 mM methionine, 1 mM glutamate (lysis buffer) supplemented with 5 mM MgSO4,

1 mM PEFA-Bloc (Roche), 25 µM ml$^{-1}$ DNAse I and 50 µM ml$^{-1}$ lysozyme. The His$_8$-tagged GluA2 LBD containing lysate was clarified by centrifugation at 53,000 g for 50 min and loaded onto 5 ml HisTrap HP column (GE Healthcare) pre-equilibrated in lysis buffer. After successive washing steps with lysis buffer supplemented with 25 mM imidazole, the protein was eluted with an imidazole gradient from 25 to 500 mM. GluA2 LBD containing fractions, as analysed by SDS–PAGE, were pooled an dialysed overnight at 4 °C against 20 mM Tris HCl pH 7.4, 200 mM NaCl, 5 mM methionine, 1 mM glutamate, 1 mM EDTA, 1 mM CaCl$_2$. Dialysed protein was digested with Trypsin (Sigma) at a molar ratio of 1:100 for 1 h at room temperature and stopped by the addition of 20 mM EDTA and 2 mM PEFA block. After the complete cleavage was confirmed by SDS–PAGE analysis, the protein was further purified by size-exclusion chromatography (SEC) using a Superdex 200 10/300 gel filtration column (GE Healthcare) in SEC buffer, 10 mM HEPES pH 7.0, 150 mM NaCl, 1 mM ETDA supplemented with 1 mM glutamate. The pooled protein fractions were extensively dialysed against the glutamate-free SEC buffer to get rid of glutamate. The protein was supplemented with 10 mM 5-fluorowillardiine (FW) and concentrated to 11 mg ml$^{-1}$ by a 10 kDa molecular weight cutoff Amicon ultra centrifugal device (Millipore).

**Crystallization and data collection.** GluA2 LBD_FW was crystallized at 4 °C in sitting drop using vapour diffusion. The crystal was obtained by mixing 200 µl protein-FW solution with 200 µl precipitant solution (100 mM Na/KPO$_4$ pH 6.2, 50.0% v/v PEG 200, 200 mM NaCl) using a Gryphon Robot (Art Robbins Instrument). A disk-like crystal, appearing within three days, was looped and directly flash-frozen in liquid nitrogen. Diffraction data were collected on beamline

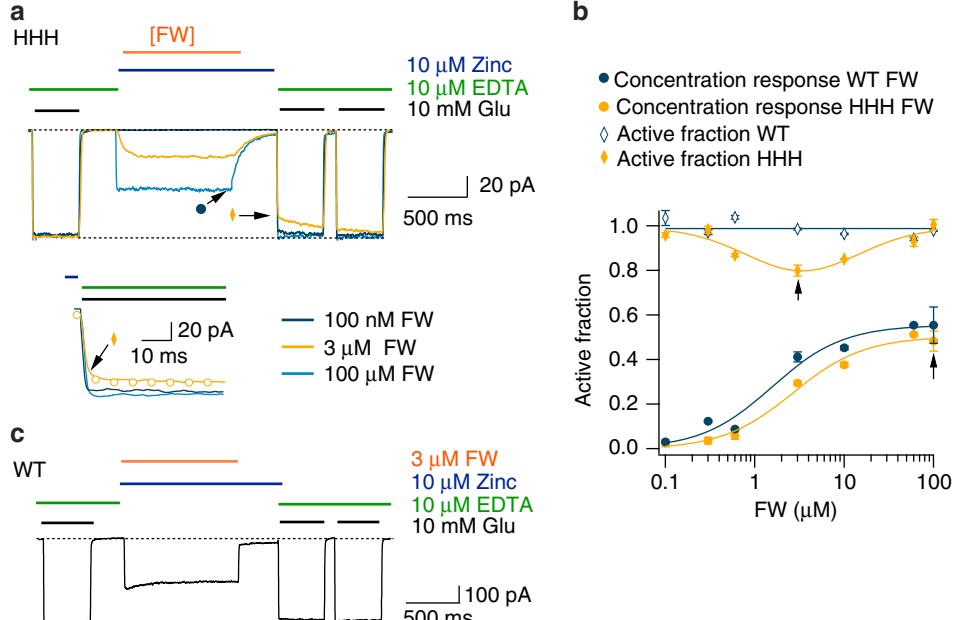

**Figure 6 | HHH traps at intermediate concentrations of FW. (a)** Trapping at different concentrations of FW. Three records from different patches showing trapping after the application of $Zn^{2+}$ (10 μM) for 100 nM (dark blue), 3 μM (orange) and 100 μM FW (light blue). The inset shows the current after trapping displaying a robust reduction occurring at 3 μM. The fits (dots) are double exponentials fitted to the recovery after trapping the arrows and symbols indicate measurements plotted in **c**. **(b)** Currents from WT GluA2 were not modified at any concentration of FW. **(c)** Dose–response curves for FW in $Zn^{2+}$ and 100 μM CTZ for WT GluA2 ($EC_{50} = 1.7 \pm 0.2$ μM) (closed blue circles) and the mutant HHH (closed yellow circles) ($EC_{50} = 2.7 \pm 0.3$ μM). The active fraction for WT GluA2 is flat (blue open dots). The active fraction for the HHH mutant (open yellow circles) is fit by a log-normal function with a maximum inhibition at 3 μM FW. All error bars are s.e.m. ($n = 5$).

BL 14.1 at the BESSY II electron-storage ring, Helmholtz Zentrum Berlin für Materialien und Energie. Data were processed using XDSapp[41].

**Phasing and refinement.** The GluA2 LBD_FW crystal indexed in space group $P2_12_12$ and diffracted to 1.23 Å. The structure was solved by molecular replacement using Phaser-MR[42] from the CCP4 Suite and the GluA2 LDB in complex with glutamate (PDB 1FTJ, chain A) as a search model[43]. Iterative cycles of model building in Coot[44] and Phenix[45] led to a model of GluA2 LBD_FW with an $R_{cryst}$ and $R_{free}$ of 12.1 and 14.7, respectively. $R_{free}$ was calculated using 5% of the reflections, which we omitted at random from the refinement. The final model includes one molecule in the asymmetric unit (from Gly1 to Gly264). Structure visualization and analysis were performed in PyMOL. Data collection and refinement statistics are summarized in Table 1.

**Primers and plasmids.** The mutants for functional studies were generated by overlap PCR on the GluA2flip template (GI: 8393475) in the pRK5 vector. The cDNA encoded a Q at the Q/R editing site. The cysteine mutations at positions 664, 665 and 666 were inserted by the following primer pairs:
Fw664 (5′-AGGAGATCTAAATGTGCAGTGTTTGATAAAATG-3′)
Rw664: (5′-ATCAAACACTGCACATTTAGATCTCCTG-3′);
Fw665: (5′-AGATCTAAAATCTGTGTGTTTGATAAAATG-3′),
Rw665: (5′-TTTATCAAACACACAGATTTTAGATCTC-3′);
Fw666 (5′-TCTAAAATCGCATGTTTTGATAAAATG3′),
Rw666 (5′-CATTTTATCAAAACATGCGATTTTAGATC-3′).
The T1 site (D668H, T672H, K761H) was generated with the following primer pairs:
Fw668,672: (5′-GTGTTTCACAAAATGTGGCACTATATGAGG-3′),
Rw668,672: (5′-CCTCATATAGTGCCACATTTTGTGAAACAC-3′)
Fw761: (5′-GTCTTAGACCACCTGAAAAAC-3′)
Rw761: (5′-GTTTTTCAGGTGGTCTAAGAC-3′).
The HH site (D668H, K765H) was engineered using following primer pairs:
Fw668 (5′- GCAGTGTTTCACAAAATGTGG-3′)
Rw668: (5′-CCACATTTTGTGAAACACTGC-3′)
Fw765: (5′-GACAAGCTGAAAAACCACTGGTGG-3′).
Rw765: (5′-CCACCAGTGGTTTTTCAGCTTGTC-3′).
The HHH mutant (G437H, K439H, D456H) was generated using the following primer pairs:
Fw437,439: (5′-GCCAAACACTGTCACTTCCACTACAAGCTG-3′)
Rw437,439: (5′-CAGCTTGTAGTGGAAGTGACAGTGTTTGGC-3′),

Fw456 (5′-CAGGGATGCCCACACCAAAATTTGGAATGG-3′)
Rw456 (5′-CCATTCCAAATTTTGGTGTGGGCATCCCTG-3′).
For each mutant, the entire amplified cassette was confirmed by double-stranded DNA sequencing.

**Electrophysiology.** Wild type and mutant AMPA receptors were expressed transiently in HEK-293 cells for outside-out patch recording. All patches were voltage clamped between $-30$ and $-60$ mV. Currents were filtered at 1–10 kHz ($-3$ dB cutoff, 8-pole Bessel) and recorded using Axograph X (Axograph Scientific) via an Instrutech ITC-18 interface (HEKA) at 20 kHz sampling rate.

The external solution in all experiments contained 150 mM NaCl, 0.1 mM $MgCl_2$, 0.1 mM $CaCl_2$, 5 mM HEPES, titrated to pH 7.3 with NaOH, to which added different drugs. For metal bridging experiments, $Zn^{2+}$ was included at 10 μM in the external solution. To achieve $Zn^{2+}$ free conditions, EDTA (10 μM), a potent $Zn^{2+}$ chelator ($K_D^{Zn2+} = 10^{-16.4}$ M), was added to the external solution. CTZ stock solution was prepared in DMSO and added at 100 μM to the external solution. Drugs were obtained from Tocris Bioscience, Ascent Scientific or Sigma Aldrich.

To measure the state dependence of trapping in kainate and other partial agonists in the presence of $Zn^{2+}$, we determined the baseline for activation by 10 mM glutamate in the presence of 10 μM EDTA, followed by application of $Zn^{2+}$ (10 μM) and the chosen agonist via the third barrel of the perfusion tool, for 1 s. Willardiine agonists unbind relatively slowly GluA2, so following modification, we held the patch in a fourth solution of 10 μM zinc alone, to allow the partial agonist to unbind but to preserve any bridges that had formed. The appropriate delay for agonist unbinding was determined for each agonist on wild-type receptors (Supplementary Fig. 4). In separate experiments, to provoke wild-type and the L2 mutant receptor into distinct proportions of the resting, intermediate and active states, we co-applied different concentrations of Glu (in the presence of 100 μM CTZ) with zinc. For all experiments, we quantified the effect of trapping by determining the fraction activated by 10 mM glutamate in EDTA (10 μM) immediately after trapping as the active fraction (AF). We assessed the effects of modification by calculating the active fraction AF as follows:

$$AF = 1 - \frac{I_{slow}}{I_{pre}} \qquad (1)$$

where $I_{slow}$ was the amplitude of the slow component of the double exponential fit to the current immediately following CuPhen treatment and $I_{pre}$ was the peak current before treatment[22]. The current was fit with a double exponential. We took the fast component to be activation by glutamate, and the slow component to

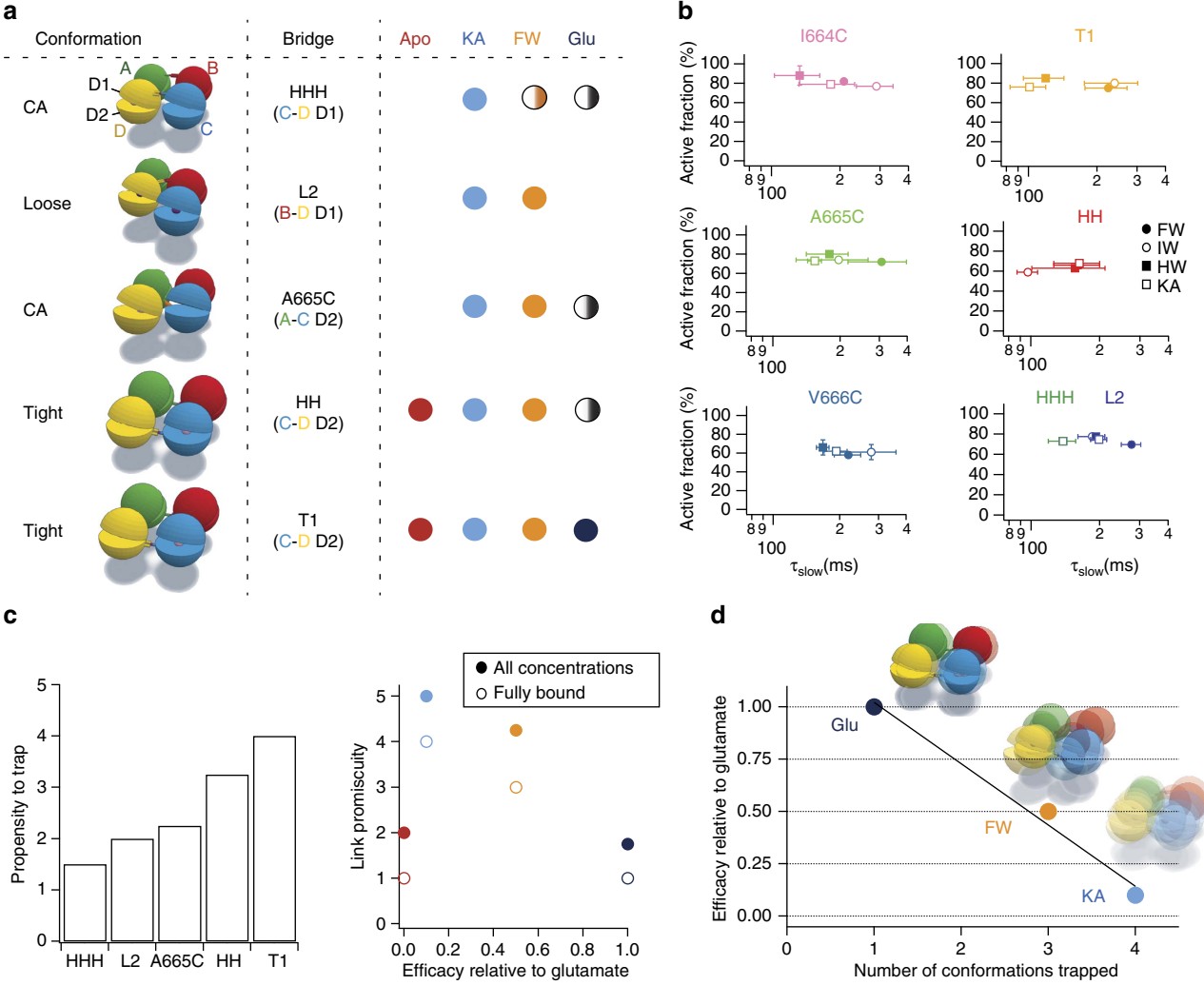

**Figure 7 | The promiscuity of cross-linking is inversely correlated with agonist efficacy.** (**a**) The table presents the cartoons of the closed angle[22], Loose and Tight conformations[23], showing the localization of the engineered bridges that either crosslink the upper D1 or the lower D2 lobe of the individual subunits (A in green, B in red, C in blue and D in yellow). The entries in the table indicate observed crosslinks for the different trapping mutations that were tested in the presence of KA, FW and glutamate. Full circles indicate trapping at saturating concentrations, whereas, half-filled circles indicate trapping at intermediate concentrations of the respective agonist. (**b**) Trapping profiles for all the mutants tested (different colours) in different partial agonists. Each plot compares the active fraction versus the time constant of recovery from trapping ($\tau_{slow}$). All error bars are s.e.m. ($n = 5$). (**c**) Each bridge has a different propensity to trap a range of distinct conformations (left panel). The crosslinking promiscuity (right panel), assessed over all agonist concentrations (filled symbols) or only for fully-bound LBD tetramers (open symbols) is plotted in relation to efficacy. A crosslink in saturating ligand was scored 1 unit, and in partially occupied LBD layers, a crosslink was scored as 0.25 units, because of the greater conformational variation in the latter case. (**d**) Cartoon illustrating the proposed mechanistic relation between conformational ensembles, efficacy and the number of different states trapped for the three classes of agonist.

represent the breaking of any bridges that formed. The amplitude of the slow component was the trapped fraction, and we subtracted this fraction from 1 to get the active fraction. A log-normal function was fitted to the active fraction of the L2 mutant activated by FW. Disulfide crosslinking of cysteine mutants was performed in a similar way, with patches being modified by CuPhen (10 μM) instead of $Zn^{2+}$, and being otherwise held in DTT (1 mM) rather than EDTA. Following modification in the presence of willardiines, the patches were held in a CuPhen-containing solution for agonist to unbinding for the same period as with experiments with $Zn^{2+}$.

We measured concentration–response curves for wild-type and mutant receptors in the presence of 100 μM CTZ to block desensitization. We obtained the $EC_{50}$ and maximum extent of activation relative to glutamate from fits to the Hill equation,

$$\frac{I}{I_{max}} = \frac{[A]^n}{[A]^n + [EC_{50}]^n} \qquad (2)$$

where $n$ is the Hill coefficient, $I_{max}$ is the maximum response and $[A]$ stands for the agonist concentration.

All $p$-values were determined by a nonparametric randomization test (using $\geq 10^5$ iterations; DCPyPs suite, https://code.google.com/p/dc-pyps/). A paired randomization test was used, where the same patch was compared between different conditions. The spread of the data is indicated as s.d. of the mean.

**Data availability.** The gene GRIA2 (from *Rattus Norvegicus*) was used in this work. The coordinates and structure factors for the FW-bound LBD structure were deposited in the Protein Data Bank (PDB) under accession code 5JEI. The other data that support the findings of this study are available from the authors on reasonable request.

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

## Acknowledgements

This work was supported by the ERC grant 647895 'GluActive', the Deutsche Forschungsgemeinschaft (DFG) grant PL619.1, and the DFG Cluster of Excellence Neuro-Cure (EXC-257) grant to A.P. H.S. was the recipient of a Long-Term Fellowship from Human Frontier Science Program (HFSP). C.E. was the recipient of an Erwin-Schrödinger Postdoctoral Fellowship (J3682-B21) of the Austrian Science fund (FWF). X-ray data collection was performed at the Joint MX laboratory, Helmholtz-Zentrum Berlin fur Materialien und Energie, Elektronenspeicherring BESSY II, Berlin, Germany. We thank Yvette Roske and the BESSY staff for their support.

## Author contributions

H.S. and C.E. designed and performed experiments, analysed the data and wrote the paper. M.C. performed experiments, analysed the data. A.P. designed experiments, analysed the data and wrote the paper.

## Additional information

**Competing financial interests:** The authors declare no competing financial interests.

