## [Peer Review File · Nature Communications]

Reviewer #1 (Remarks to the Author)

Summary of the key results

This ms reports a new high resolution structure of the ligand-binding layer of the GluA2 tetramer (GluA2-LBD)₄ in complex with the partial agonist FW.

Structures for the (GluA2-LBD)₄ ensemble were previously reported in complex with the AMPA receptor antagonist CNQX (Neuron 2013) and subsequently with the natural AMPA receptor agonist Glu (Biophys J 2016). These two preceding reports suggested for the first time the existence of a more compact arrangement of the LBD layer (denoted AC) relative to the arrangement proposed by Sobolevsky et al for the LBD layer of a functional tetrameric receptor (Nature 2009), which appears more extended (here denoted AO). Structure-based mutagenesis coupled with current recordings supported the hypothesis that the AC conformation can be populated during physiologic activation of the AMPA receptor (with Glu). The significance of this work emanated from the current dearth of structural data on tetrameric AMPA receptors; In the current ms authors extend this previous work on the LBD layer by presenting a new structure for a tetrameric LBD protein, with FW molecules complexing the ligand binding site. The authors go on to compare this new FW- LBD₄ structure with the previously published Glu-LBD₄ (BJ 2016) and with the Sobolevsky model of a functional receptor, and interpret the differences as indicative of a general mechanism by which partial agonist elicit less current from AMPARs than from the full agonist Glu. This interpretation is presented in contrast to the initial model of partial agonism at AMPA receptors put forth by the pioneering studies of Jin with Gouaux in 2003.

However, the field has evolved quite a bit since 2003, and the differences reported here between the FW- and Glu-bound structures fit well within current theories of partial agonism at AMPARs, which take into consideration both differences in the extent of closure and in the dynamics of the clamshell. The initial hypothesis of Jin with Gouaux proposed that the extent of LBD closure around an agonist was the principal determinant to agonist efficacy. Subsequent studies with LBD monomers and dimers, with crystallography coupled with functional measurements have added layers of complexity, which include possible twisting motions between the two LBD lobes, differences in the degrees of conformational freedom, etc.

Originality and interest: if not novel, please give references

This report illustrates for the first time differences in the arrangement of LBD dimers within the isolated LBD tetrameric layer, when the binding sites are occupied by a partial agonist FW relative to Glu-occupied protein. The new structure represents an important data point towards the correct understanding of the wide array of AMPA receptor conformations that support its function; structural biologists interested in possible AMPA receptor conformations will very likely to read and cite the paper.

Data & methodology: validity of approach, quality of data, quality of presentation

The methodology is rigorous; in addition, appropriate controls and functional data support the hypothesis that the observed arrangement of the separated LBD layer can occur during AMPA receptor activation.

Conclusions: robustness, validity, reliability

Conclusions are wildly exaggerated; results are interpreted as a general mechanism for partial agonist; this is overstated; the structure is of a detached layer, not of an entire receptor and this limitation is not clearly articulated; only one partial agonists structure is shown, and others may work differently;

The expectation that one can derive a 'general' mechanism of partial agonism is contradicted by many studies that illustrate clear differences among LBD ligands in the residues they contact, the position in which they bind, and the kinetic changes they produce in receptor activation (see Nowak, Howe, Traynelis, Oswald, etc)

Suggested improvements:

Ms is more suitable to a more specialized journal

References: appropriate credit to previous work?

Within the limited scope of structural investigations reported here, references are appropriate; functional aspects of receptor activation and partial agonism are ignored here and in previous

papers on LBD layer (Neuron 2013, BJ 2016). For example, see work by Nowak and colleagues.
Clarity and context: lucidity of abstract/summary, appropriateness of abstract, introduction and conclusions

The paper claims to address the general question of partial agonism; instead it describes differences between two structures, one with Glu and one with FW; The scholarship is weak in the context of considering mechanisms of partial agonism in ion channels;

The narrative is structure-centric thus limiting the audience for which this ms will be of use. electrophysiologists may be offended by dated terminology and apparent unfamiliarity or unawareness of literature on functional aspects of activation mechanisms (see work by the Howe and Nowak labs on activation sequence); neuroscientists may be put off by statements that strictly define AMPARs as postsynaptic (extensive literature illustrates their much broader distribution); pharmacologists will look down at defining partial agonists as weak

Significance

The importance of understanding how partial agonists produce responses that are smaller than those produced by natural agonists has practical and theoretic aspects; the practical interest arises from the observation that some partial agonists may be more effective drugs than the full agonist; they are also used as tools to investigate the activation mechanisms of neurotransmitter receptors and to develop and test increasingly realistic models of receptor activation. Within this context a major weakness is that the ms assumes that all partial agonists work by the same mechanism and puts forth blanket statements about partial agonist mechanisms based on behaviors observed with only one partial agonist, FW. Functional data accumulating from AMPAR, but also AChR and NMDAR obtained with series of agonists of varying efficacies have clearly demonstrated agonist-specific ways in which receptor activity is modified relative to the natural agonist.

Reviewer #2 (Remarks to the Author)

This is a very interesting report by Salazar et al. on the conformational ensembles of the AMPA receptor in the presence of various partial agonists. The conclusions (i.e., partial agonists have a larger conformational ensemble than full agonists) are largely consistent with previous reports, but the use of the tetrameric LBD provides new and interesting insights on the presumed quaternary structures. I have a few relatively minor suggestions for the authors to consider:

1) I think it might be a challenge for most readers (including me) to follow the arguments spatially without better labeling of the figures. At minimum, I would label A665 and L748 in Figure 1D and then provide a figure that shows the positions of all of the mutants in Table 1.

2) Line 364: What is "W"?

3) Line 433: "For some reason, all bridges we have studied are inhibitory..." I'm not totally sure what this means. Are they saying that the mutations alone are inhibitory or that when a bridge is formed due to the mutation, the result is inhibitory? For the former, I would agree with the "for some reason" clause. For the latter, I would think that this would be expected. The trapping in most cases would be in state that would not necessarily be conducive to channel opening.

4) I think the results with kainate are interesting. In terms of the monomeric LBDs, certainly the binding of willardiines results in more conformational flexibility than the binding of glutamate. In terms of kainate, it too results in more conformational flexibility than glutamate, but it not necessarily more than the willardiines (probably due to the additional interactions in the binding site). Obviously this might not translate to the crosslinks described in this study.

5) I suppose it is not surprising that fifty-year-old models (MWC and KNF) of allosteric enzymes may not be able to capture the dramatic increase in conformational details that we can now obtain from neurotransmitter receptors. If anything, the results are more consistent with the general model proposed by Hammes & Wu (Ann. Rev. Biophys. Bioeng., 1974) only forty-years-ago.

Reviewer #3 (Remarks to the Author)

This manuscript describes complementary crystallographic and electrophysiological studies on the AMPA receptor LBD. The authors contemplate mechanisms of partial agonism and use their very high resolution structure of the LBD in complex with the partial agonist FW, along with several other published structures, to design mutants and test them in fast perfusion patch clamp assays. The overall finding is that partial agonists have limited efficacy because they stabilize non-activating conformations in addition to the activating conformation. Further, the authors propose that the number of non-productive conformations that a partial agonist stabilizes correlates inversely with its efficacy. My impression is that the approach is elegant, the idea is compelling and the results likely support the conclusions, but the data, especially related to structures and the rationale based on structures, is not presented clearly enough to allow the reader to easily interpret the mechanistic ideas and results. It is also a bit unclear to me if this study represents a major advance beyond what was done in the two previous crosslinking studies from the same group (references 18,19). I make specific suggestions below.

Major:

1. Clarity to a non-iGluR aficionado could be improved dramatically by including more orienting information on receptor structure, information that would aid in background for all the analysis described in the results. For example, please label ATD, LBD and TMD in Fig 1a. As another example, Fig 1d legend states that its right panel is oriented as full length structures presented in Panels A and B. To me it looks like that can't be right-it looks like it is rotated by 90 degrees. Also this panel is so small that I cannot see the ligand in grey spheres. Please examine manuscript for confusing points like this (confusing to someone not intimately familiar with all recent high resolution structures of isolated AMPA receptor domain structures).
2. The crystal structure contains a monomer in the ASU, but crystal lattice contacts and packing in the unit cell are used, from what I understand, to infer how the subunits arrange in the native tetrameric receptor. Is that a realistic assumption? It makes sense that ligand binding would affect conformation of the monomer to which it binds. It does not make intuitive sense to me that the crystal packing would then be affected in a way that has any underlying physiological meaning related to full length receptor structure. Additionally, please compare your hypothetical tetramer based on crystal packing to the arrangement of LBD subunits seen in full-length structures.
3. Would 10 mM FW not be expected to potently stabilize the desensitized state in the context of the equilibrium environment of crystallization? How are you certain that you are examining a structure relevant to the open state and not the desensitized state? The assertion, unreferenced that intact D1 interfaces indicate that the structure is not related to desensitization (lines 240-241) is not convincing to me. For example, the PMID 25103407 paper (2014 Sobolevsky lab in Science) obtained a full-length AMPA receptor structure with another willardiine agonist in conditions that should stabilize the desensitized state and found the D1-D1 interface to be intact.

Minor:

1. The description of angles used to compare different structures (lines 244-266) is confusing without visual aids; please add a Figure, could be supplementary, showing side-by-side comparisons of the structures discussed with the angles shown.
2. Is discussion of pre-open or flipped states relevant, perhaps in the intro or discussion (Nature. 2008 Aug 7;454(7205):722-7)?
3. The statement "... is a function of both time and space" is confusing to me. Maybe say "conformational space" to be more specific.

Reviewers' comments:

Reviewer #1 (Remarks to the Author):

Summary of the key results

This ms reports a new high resolution structure of the ligand-binding layer of the GluA2 tetramer (GluA2-LBD)₄ in complex with the partial agonist FW.

Structures for the (GluA2-LBD)₄ ensemble were previously reported in complex with the AMPA receptor antagonist CNQX (Neuron 2013) and subsequently with the natural AMPA receptor agonist Glu (Biophys J 2016). These two preceding reports suggested for the first time the existence of a more compact arrangement of the LBD layer (denoted AC) relative to the arrangement proposed by Sobolevsky et al for the LBD layer of a functional tetrameric receptor (Nature 2009), which appears more extended (here denoted AO). Structure-based mutagenesis coupled with current recordings supported the hypothesis that the AC conformation can be populated during physiologic activation of the AMPA receptor (with Glu). The significance of this work emanated from the current dearth of structural data on tetrameric AMPA receptors;

In the current ms authors extend this previous work on the LBD layer by presenting a new structure for a tetrameric LBD protein, with FW molecules complexing the ligand binding site.

The authors go on to compare this new FW- LBD₄ structure with the previously published Glu-LBD₄ (BJ 2016) and with the Sobolevsky model of a functional receptor, and interpret the differences as indicative of a general mechanism by which partial agonist elicit less current from AMPARs than from the full agonist Glu. This interpretation is presented in contrast to the initial model of partial agonism at AMPA receptors put forth by the pioneering studies of Jin with Gouaux in 2003.

However, the field has evolved quite a bit since 2003, and the differences reported here between the FW- and Glu-bound structures fit well within current theories of partial agonism at AMPARs, which take into consideration both differences in the extent of closure and in the dynamics of the clamshell. The initial hypothesis of Jin with Gouaux proposed that the extent of LBD closure around an agonist was the principal determinant to agonist efficacy. Subsequent studies with LBD monomers and dimers, with crystallography coupled with functional measurements have added layers of complexity, which include possible twisting motions between the two LBD lobes, differences in the degrees of conformational freedom, etc.

We are grateful for the nuance of these comments. We have revised the manuscript to pay more appropriate attention to studies that have appeared in the interval since 2003. We added a paragraph to the discussion that we feel gives proper prominence to the work on the level of individual subunits, and the relation to single channel data.

Originality and interest: if not novel, please give references

This report illustrates for the first time differences in the arrangement of LBD dimers within the isolated LBD tetrameric layer, when the binding sites are occupied by a partial agonist FW relative to Glu-occupied protein. The new structure represents an important data point towards the correct understanding of the wide array of AMPA receptor conformations that support its function; structural biologists interested in possible AMPA receptor conformations will very likely to read and cite the paper.

We are grateful for these positive comments. Thank you.

Data & methodology: validity of approach, quality of data, quality of presentation
The methodology is rigorous; in addition, appropriate controls and functional data support the hypothesis that the observed arrangement of the separated LBD layer can occur during AMPA receptor activation.

Thank you for this kind assessment.

Conclusions: robustness, validity, reliability

Conclusions are wildly exaggerated; results are interpreted as a general mechanism for partial agonist; this is overstated; the structure is of a detached layer, not of an entire receptor and this limitation is not clearly articulated; only one partial agonist's structure is shown, and others may work differently;

The expectation that one can derive a 'general' mechanism of partial agonism is contradicted by many studies that illustrate clear differences among LBD ligands in the residues they contact, the position in which they bind, and the kinetic changes they produce in receptor activation (see Nowak, Howe, Traynelis, Oswald, etc)

Although we only solved one structure with one partial agonist, we have tested a panel of agonists. Not only did we find that previous reported efficacy differences between willardiine agonists were perhaps overstated (Jin et al 2003 paper), we also examined kainate which is a partial agonist with truly distinct efficacy. Whilst the structure is from an isolated part of the receptor, we feel that our extensive crosslinking results obtained during functional activation in a lipid membrane of a mammalian cell are quite distinct from crystallising the full-length channel or doing cryoEM in detergent. We think that the predictive power of our structure to identify unique crosslinking conditions (for the L2 bridge) validate working in this way.

All the measurements we present here have little to do with kinetics. We are looking at equilibrium (or perhaps steady state). Mechanisms of kinetic effects on activation might be more complex still - we do not try to address this point.

We would like to make a single specific rebuttal of the claim that we do not obtain general insight. If partial agonism was mainly about the chemistry of the binding site, or binding modes, then the "intermediate-FW" experiment (Figure 5) ought to have failed. If it's about chemistry, tetramers partly bound by FW would have reduced activity because of particular, within-subunit conformational changes. The fact that partially-FW bound receptors access the same conformation as a KA-saturated tetramer (one not available to resting, FW or Glu-saturated tetramers) shows that the tetramer arrangement of subunits is an important factor in activity. The spectrum of tetramer arrangements probably derive from different LBD stabilities and conformations, and we have sought to underline this point, giving credit to studies on this topic, in a new paragraph in the discussion (page 18). What we show (and what is new) is that the ensemble of subunits arrangements, detected by intersubunit bridges, is a strong determinant of activity.

Suggested improvements:

Ms is more suitable to a more specialized journal

This point is hard to answer, but the topic is undoubtedly of high general interest (Jin et al 2003 paper has 300+ citations). It was recently brought to our attention that a related mechanism has been proposed for G-protein coupled receptors:

Ligand Binding Ensembles Determine Graded Agonist Efficacies at a G Protein-Coupled Receptor Bock A., Bermudez M., Krebs F., Matera C., Chirinda B., Sydow D., Dallanoce C., Holzgrabe U., De Amici M., Lohse M. J., Wolber G. and Mohr K.J Biol Chem (2016)

References: appropriate credit to previous work?

Within the limited scope of structural investigations reported here, references are appropriate; functional aspects of receptor activation and partial agonism are ignored here and in previous papers on LBD layer (Neuron 2013, BJ 2016). For example, see work by Nowak and colleagues.

We are happy to cite relevant work on this topic. We did not discuss partial agonism in detail in our previous work. Indeed, partial agonists induce modal gating in AMPA receptors and we added a paragraph that includes reference to the single channel data reported by Nowak and colleagues. However, the mechanism of the modal gating remains unclear, limiting the extent to which we can relate it to our measurements on ensemble currents.

Clarity and context: lucidity of abstract/summary, appropriateness of abstract, introduction and conclusions

The paper claims to address the general question of partial agonism; instead it describes differences between two structures, one with Glu and one with FW; The scholarship is weak in the context of considering mechanisms of partial agonism in ion channels; The narrative is structure-centric thus limiting the audience for which this ms will be of use. electrophysiologists may be offended by dated terminology and apparent unfamiliarity or unawareness of literature on functional aspects of activation mechanisms (see work by the Howe and Nowak labs on activation sequence); neuroscientists may be put off by statements that strictly define AMPARs as postsynaptic (extensive literature illustrates their much broader distribution); pharmacologists will look down at defining partial agonists as weak

Whilst we can accept that one can always improve a manuscript with clearer writing, some of these criticisms are hard to fathom.

We discuss quite a few structures in the paper, but report only one. We do not feel that the manuscript is structure-centric, because we talk about activity of the receptor a great deal. However, to address this concern, we have added a section to the discussion about previous work that dealt with activation (see below). These works are generally speculative about mechanism - they don't test the mechanisms that they propose to rationalise their observations.

We are not aware that our terminology was out of date. Because no example is given, it's very challenging to respond to this point.

We want to cite work in a fair way. The work suggested from Nowak and Howe is in the context of the monomer or dimer of subunits, but does include single channel recording in some instances. This valuable work addresses clamshell closure and stability. We are of course well aware of these studies, which generated the idea that clamshell stability is a major factor in partial agonism, and that we mentioned on line 575:

“Taken together, the poor ability of partial agonists to stably close the LBD clamshell, produces increasingly profound conformational promiscuity.”

We also cited the work from Howe’s lab on the mutations at the lips of the clamshell in the introduction (line 78). We apologise for our carelessness in omitting other citations. We now cite papers from Nowak and Oswald here, and include a section in the discussion to discuss previous work on partial agonist activation. However, making sense out of binding site chemistry, and relating it to partial agonism via GluA3 single channel gating modes, remains challenging. Beyond plausibility, we would contend that test cases, such as kainate on the D655A mutant, are difficult to rationalise. This mutation increases KA efficacy (Holley et al Biochemistry 2012) even though the peptide flip and consequent participation of D655 in a salt bridge network between binding domain lobes were previously taken to be major determinants of high-activity (Poon et al MolPharm 2011). Clearly work is needed to make this link more concrete, which limits the extent to which we can relate these studies to our work.

We did not define AMPARs as purely postsynaptic, we merely described how postsynaptic receptors are activated (line 46). One could argue that the broad extra synaptic distribution of AMPA receptors is merely a consequence of their delivery at extrasynaptic sites and has little role in brain function. But such a discussion is outside our purview.

Weak seemed to us a perfectly reasonable description of a partial agonist that is as bad at activating GluA2 as Kainate is - it is not our intention to offend anybody.

Significance

The importance of understanding how partial agonists produce responses that are smaller than those produced by natural agonists has practical and theoretic aspects; the practical interest arises from the observation that some partial agonists may be more effective drugs than the full agonist; they are also used as tools to investigate the activation mechanisms of neurotransmitter receptors and to develop and test increasingly realistic models of receptor activation. Within this context a major weakness is that the ms assumes that all partial agonists work by the same mechanism and puts forth blanket statements about partial agonist mechanisms based on behaviors observed with only one partial agonist, FW. Functional data accumulating from AMPAR, but also AChR and NMDAR obtained with series of agonists of varying efficacies have clearly demonstrated agonist-specific ways in which receptor activity is modified relative to the natural agonist.

As pointed out by the other referees, we have worked on various partial agonists. Although we were only able to solve one relevant structure, we obtained rich information from full-length receptors using a broad panel of engineered crosslinks. Therefore, we are basing our ideas on, at the very least, 4 distinct agonists (KA, W, FW, and Glu).

We apologise if we have given the impression that we feel our mechanism was more general than it is, and can be extended to other receptor families. This was not our intention. Therefore, we wrote in the title that the work refers to AMPA-type glutamate

receptors. We assumed it to be clear that we are referring to AMPA receptors throughout the paper. Indeed, mechanisms in CLRs and NMDARs are likely different - we discussed this point in the paragraph beginning around line 567.

The different mechanisms of partial agonism are also discussed in the recent review "Structural mechanisms of activation and desensitization in neurotransmitter-gated ion channels" in NSMB.

We have altered the text in several places (lines 30, 56, 106, 270, 341, 614) to make it more clear that we are referring to the AMPA Receptor.

Reviewer #2 (Remarks to the Author):

This is a very interesting report by Salazar et al. on the conformational ensembles of the AMPA receptor in the presence of various partial agonists. The conclusions (i.e., partial agonists have a larger conformational ensemble than full agonists) are largely consistent with previous reports, but the use of the tetrameric LBD provides new and interesting insights on the presumed quaternary structures.

Thank you very much for these supportive comments and the recognition of the new insights we have been able to present.

I have a few relatively minor suggestions for the authors to consider:

- 1) I think it might be a challenge for most readers (including me) to follow the arguments spatially without better labeling of the figures. At minimum, I would label A665 and L748 in Figure 1D and then provide a figure that shows the positions of all of the mutants in Table 1.

Thank you for this helpful suggestion. We have improved the presentation of figure 1 by labelling the domains of the full-length structures and indicating the position of L748 and A665. The positions of all of the mutants are shown within the LBD structures and schematically in cartoons in figure 1, 3 and 4. We decided to not duplicate this information in Table 1, but instead refer explicitly to these cartoon representations in the table legend and the main text.

- 2) Line 364: What is "W"?

We apologise for the confusion arising from our inconsistent nomenclature. We changed the wording here to make it clear that we are referring to willardiine (HW).

- 3) Line 433: "For some reason, all bridges we have studied are inhibitory..." I'm not totally sure what this means. Are they saying that the mutations alone are inhibitory or that when a bridge is formed due to the mutation, the result is inhibitory? For the former, I would agree with the "for some reason" clause. For the latter, I would think that this would be expected. The trapping in most cases would be in state that would not necessarily be conducive to channel opening.

Thank you for addressing this subtle point. We have re-written the sentence to indicate more clearly that we intended the latter - that bridges formed are generally inhibitory.

We are genuinely surprised that we have found no lock-open or potentiating bridges in our studies of LBD crosslinking. However, we cannot exclude the possibility that we just looked in the wrong places. We added text to the discussion (around line 484) to note that bridges can have a lock-open effect in some other channels.

Less conclusively, crosslinks between LBDs in an active dimer appear potentiate the steady-state current due to the receptor (Weston et al, 2006 NSMB), by blocking desensitization, but this point is controversial [see Daniels et al 2013 JPhysiol]. [redacted]

For these reasons, we prefer not to state that inhibition is expected.

4) I think the results with kainate are interesting. In terms of the monomeric LBDs, certainly the binding of willardiines results in more conformational flexibility than the binding of glutamate. In terms of kainate, it too results in more conformational flexibility than glutamate, but it not necessarily more than the willardiines (probably due to the additional interactions in the binding site). Obviously this might not translate to the crosslinks described in this study.

Thank you for this point. We now include references to the phenomena suggested.

Partial agonist like iodowillardine (IW), crystals show a wide range of lobe closures ⁽¹⁾and there are large scale dynamics in the NMR spectra³. In the case of kainate an apparent steric clash of the isoprenyl group of kainate with the side chain of L650 in GluA2 produces a block of the closure of the clamshell. Despite this steric hindrance, disulfide trapping experiments show that fully closed form of the LBD of GluA2 can be obtained in the presence of the partial agonists IW and kainate ⁴.

We now mentioned these observations more explicitly in the discussion in a new paragraph around line 567. We also point out that the extent of differences between the dynamics of LBDs bound by kainate and willardiines remains unclear.

One final point - the domain closure in isolated LBDs differs from that seen in full-length structures. Our technique of measuring geometry during activation of receptors in membranes circumvents this limitation.

*1. Jin, R. & Gouaux, E. Probing the function, conformational plasticity, and dimer-dimer contacts of the GluR2 ligand-binding core: studies of 5-substituted willardiines and GluR2 S1S2 in the crystal. *Biochemistry* **42**, 5201–13 (2003).*

*3. Fenwick, M. K. & Oswald, R. E. *NIH Public Access*. **378**, 673–685 (2009).*

*4. Ahmed, A. H., Wang, S., Chuang, H.-H. & Oswald, R. E. Mechanism of AMPA receptor activation by partial agonists: disulfide trapping of closed lobe conformations. *J. Biol. Chem.* **286**, 35257–66 (2011).*

5) I suppose it is not surprising that fifty-year-old models (MWC and KNF) of allosteric enzymes may not be able to capture the dramatic increase in conformational details that we can now obtain from neurotransmitter receptors. If anything, the results are more

consistent with the general model proposed by Hammes & Wu (Ann. Rev. Biophys. Bioeng., 1974) only forty-years-ago.

Thank you for this interesting point. We added the suggested reference in line 70 pointing out that these models are edge cases of a more general model that includes intermediates. However, the Hammes and Wu conception is in any case likely still too simplified because it only considers two conformations per subunit. Life seems more complicated now than models derived in the 1970s might suggest, prescient though the paper in question was.

Reviewer #3 (Remarks to the Author):

This manuscript describes complementary crystallographic and electrophysiological studies on the AMPA receptor LBD. The authors contemplate mechanisms of partial agonism and use their very high resolution structure of the LBD in complex with the partial agonist FW, along with several other published structures, to design mutants and test them in fast perfusion patch clamp assays. The overall finding is that partial agonists have limited efficacy because they stabilize non-activating conformations in addition to the activating conformation. Further, the authors propose that the number of non-productive conformations that a partial agonist stabilizes correlates inversely with its efficacy. My impression is that the approach is elegant, the idea is compelling and the results likely support the conclusions, but the data, especially related to structures and the rationale based on structures, is not presented clearly enough to allow the reader to easily interpret the mechanistic ideas and results. It is also a bit unclear to me if this study represents a major advance beyond what was done in the two previous crosslinking studies from the same group (references 18,19). I make specific suggestions below.

We thank the referee for these supportive comments. We apologise for the lack of clarity and have striven to improve on this point. We also try to address the question of the specific advances that this manuscript provides more clearly in the final paragraph of the introduction (around line 94).

Major:

1. Clarity to a non-iGluR aficionado could be improved dramatically by including more orienting information on receptor structure, information that would aid in background for all the analysis described in the results. For example, please label ATD, LBD and TMD in Fig 1a. As another example, Fig 1d legend states that its right panel is oriented as full length structures presented in Panels A and B. To me it looks like that can't be right-it looks like it is rotated by 90 degrees. Also this panel is so small that I cannot see the ligand in grey spheres. Please examine manuscript for confusing points like this (confusing to someone not intimately familiar with all recent high resolution structures of isolated AMPA receptor domain structures).

We are grateful for these insightful comments and have made several modifications to the figures accordingly.

We now labelled the domains of the full-length structures. For better contrast, we changed the colours of the ligands and the modulator to black and cyan, respectively. We included an arrow to explain the orientation of the tetrameric LBDs better. We also changed the arrangement and size of panels to make the relation between the side and top view more evident.

2. The crystal structure contains a monomer in the ASU, but crystal lattice contacts and packing in the unit cell are used, from what I understand, to infer how the subunits arrange in the native tetrameric receptor. Is that a realistic assumption? It makes sense that ligand binding would affect conformation of the monomer to which it binds. It does not make intuitive sense to me that the crystal packing would then be affected in a way that has any underlying physiological meaning related to full length receptor structure. Additionally, please compare your hypothetical tetramer based on crystal packing to the arrangement of LBD subunits seen in full-length structures.

The referee points out an important aspect. The tetrameric LBD arrangement is generated by crystal-packing and needed to be validated as a physiologically-plausible arrangement. This process is, in part, what we describe in this paper. What is new in this tetrameric structure is that subunits B and D are closer than we or anyone else has seen before. By engineering the L2 metal trap mutation, based on our sLBD_FW structure, we were able to demonstrate that this LBD arrangement is also present in functional full-length receptors in electrophysiological studies.

We extended our comparison of the arrangement of the LBDs in Supplementary Figure 1. We now include side views of the tetramer as well, to provide a clearer view of the various structures and how they relate to each other.

3. Would 10 mM FW not be expected to potently stabilize the desensitized state in the context of the equilibrium environment of crystallization? How are you certain that you are examining a structure relevant to the open state and not the desensitized state? The assertion, unreferenced that intact D1 interfaces indicate that the structure is not related to desensitization (lines 240-241) is not convincing to me. For example, the PMID 25103407 paper (2014 Sobolevsky lab in Science) obtained a full-length AMPA receptor structure with another willardiine agonist in conditions that should stabilize the desensitized state and found the D1-D1 interface to be intact.

We are grateful for these critical comments. The reviewer is correct, we should reference this statement and have now added references to the appropriate papers from Eric Gouaux's lab that first described the intact and broken dimers and their relation to active and desensitised states. There is a lot more data on intact D1 dimers (in other receptors and so on) but this point has been discussed at considerable length in previous work.

We very strongly disagree with the idea that the Sobolevsky lab NOW structure represents a desensitised state (as we politely pointed out at line 242; discussed in more detail in Baranovic and Plested, Biological Chemistry, 2016). This manuscript was published at the same time as several other manuscripts (and in comparative ignorance of their contents). First of all, the Sobolevsky NOW structure is very similar to partial agonist structures bound by very potent blockers of desensitisation. (Duerr et al, Cell 2014). Both Gouaux's and Mayer's labs concurrently published more compelling desensitised state structures. Indeed, Sobolevsky's structure much more likely corresponds to that of a member of the ensemble of inactive, ligand-bound, channel closed structure that we predict exists from our conception of partial agonists (as do other partial agonist structures from the Gouaux lab), even though it is distinct from our measurements in terms of geometry. Nonetheless, whilst the Sobolevsky structure remains relatively untested by complementary functional measurements, it is hard to be certain what state has been resolved. On the other hand, the trapping of our structure during functional experiments in non-desensitising

conditions, coupled to the perfectly intact active dimer interface, strongly suggests that we have captured a non-desensitised form in the crystal.

Minor:

1. The description of angles used to compare different structures (lines 244-266) is confusing without visual aids; please add a Figure, could be supplementary, showing side-by-side comparisons of the structures discussed with the angles shown.

Thank you for pointing this out. We added panels to supplementary figure 1 to address this matter.

2. Is discussion of pre-open or flipped states relevant, perhaps in the intro or discussion (Nature. 2008 Aug 7;454(7205):722-7)?

We added this citation in the introduction (line 60) and in the discussion (line 553).

3. The statement "... is a function of both time and space" is confusing to me. Maybe say "conformational space" to be more specific.

We made this change to be more specific as the referee suggested. Thank you.

Reviewer #2 (Remarks to the Author)

The authors have responded well to the comments of the reviewers. I have no additional concerns.

Reviewer #3 (Remarks to the Author)

The authors have satisfactorily addressed all my concerns.